# Genes Associated with Disturbed Cerebral Neurogenesis in the Embryonic Brain of Mouse Models of Down Syndrome

**DOI:** 10.3390/genes12101598

**Published:** 2021-10-11

**Authors:** Keiichi Ishihara

**Affiliations:** Department of Pathological Biochemistry, Kyoto Pharmaceutical University, Kyoto 607-8414, Japan; ishihara@mb.kyoto-phu.ac.jp; Tel.: +81-75-595-4656

**Keywords:** Down syndrome, prenatal neurogenesis, brain development, responsible genes, related pathway

## Abstract

Down syndrome (DS), also known as trisomy 21, is the most frequent genetic cause of intellectual disability. Although the mechanism remains unknown, delayed brain development is assumed to be involved in DS intellectual disability. Analyses with human with DS and mouse models have shown that defects in embryonic cortical neurogenesis may lead to delayed brain development. Cre-loxP-mediated chromosomal engineering has allowed the generation of a variety of mouse models carrying various partial Mmu16 segments. These mouse models are useful for determining genotype–phenotype correlations and identifying dosage-sensitive genes involved in the impaired neurogenesis. In this review, we summarize several candidate genes and pathways that have been linked to defective cortical neurogenesis in DS.

## 1. Introduction

Intellectual disability is characterized by impaired cognitive abilities, commonly defined by an intelligence quotient <70, and severe deficits in the capability to adapt to the environment and social milieu. Down syndrome (DS), caused by triplication of human chromosome 21 (Hsa21), is the most frequent genetic cause of intellectual disability. Accumulating evidence in DS individuals and DS mouse models indicates embryonic brain hypotrophy due to impaired cortical neurogenesis [1,2,3,4,5] and proliferation impairment [3,4]. Dendritic pathologies, such as a marked reduction in dendritic branching and spine density, are also reported in both DS individuals and DS mouse models [6,7]. These anomalies are thought to be key determinants for intellectual disability of DS.

According to the “gene dosage hypothesis” [8], any one of the over 400 overexpressed genes in Hsa21 [9] may contribute to the impaired neurogenesis and dendritic pathologies. Several candidate genes have recently been suggested.

In this review, we summarized the genes involved in embryonic brain hypotrophy.

## 2. Analyses of the Brains of People with DS

Although experiments with postmortem brains of human fetuses with DS have imparted very little information related to neurogenesis, immunostaining for Ki67, which is expressed in all phases of the cell cycle, demonstrated that the number of proliferating cells was markedly reduced in the embryonic hippocampus and cerebellum [4,10]. During the second trimester, reduced cellular proliferation and increased cell death result in fewer neurons in the neocortex, hippocampus, and cerebellum [4,10,11,12,13]. Fewer neurons in the ventricular zone (VZ) and subventricular zone (SVZ) suggest an underproduction of excitatory neurons, leading to an imbalance between excitatory and inhibitory neurons. Furthermore, fewer neurons and more astrocytes are found in the prenatal brain with DS, suggesting that the neural progenitor cells (NPCs) in DS show a greater shift towards glial lineages: differentiating into astrocytes, preferentially [4,13,14]. In the late gestation period, brains with DS show delayed and disorganized patterns of cortical lamination [15,16].

In addition to analyses with postmortem brains, experiments with NPCs from the fetal frontal cortex with DS also demonstrated delayed proliferation compared with those from the non-DS frontal cortex [17]. Human induced pluripotent stem cells (iPSCs) with trisomy 21 have been generated by multiple groups. Several studies have used iPSC models of DS to demonstrate defects in NPC proliferation [18], neurogenesis, the synaptic morphology/function, and the mitochondrial function [19]. Recently, an analysis of DS iPSC-derived cerebral organoids partially recapitulated the abnormalities observed in DS mouse models (see below) and postmortem DS brain samples, including a reduced proliferation rate and abnormal neurogenesis [20].

However, while postmortem studies have provided significant insights into the neuropathology of DS people, in vivo studies are necessary to understand the natural history of the human condition and how the pathology relates to neurodevelopmental outcomes. Recent advances in non-invasive imaging technologies, such as magnetic resonance imaging (MRI), have aided in our understanding of the in utero and neonatal brain development in DS [21]. In particular, advanced MRI techniques performed on living fetuses have provided an unprecedented opportunity to study the fetal brain development in cases of DS. Tarui et al. assessed the growth of fetal brains with DS using a regional volumetric analysis of fetal brain MRI, demonstrating decreased growth trajectories of the cortical plate, subcortical parenchyma, and cerebellar hemispheres in people with DS compared to controls [22].

## 3. Analyses of the Brains of Mouse Models for DS

Mouse models of DS are very useful for analyzing the DS pathophysiology in vivo. Since a large portion of Hsa21 shows synteny with the distal end of mouse chromosome 16 (Mmu16), mice carrying an extra copy of a part of Mmu16 have been generated as mouse models of DS (Figure 1).

Ts65Dn mice, which are the most well-characterized and widely used of these mice, were generated by irradiating the testes of male mice, breeding them, and screening the offspring for chromosomal rearrangements involving Mmu16 [23]. Ts65Dn mice carry an extra Mmu16 segment with roughly 90 genes syntenic to Hsa21 genes and exhibit a number of neural phenotypes affecting learning, memory, brain morphology, and synaptogenesis. Many of these effects parallel changes that are observed in individuals with DS [24]. Ts65Dn mice reportedly exhibit disturbed prenatal neurogenesis similar to that seen in individuals with DS [3]. Ts1Cje mice, which carry an unbalanced derivative—Ts(12^16^)1Cje—of a balanced translocation, were induced by gene-targeting in mouse ES cells [25]. Ts1Cje mice have an extra trisomic region coding approximately 70 genes from Scaf4 to Zbtb21 in Mmu16, which is shorter than that in Ts65Dn mice. The Ts1Cje mice also exhibit an impaired learning memory in the Morris water maze despite having a milder condition than Ts65Dn mice [26]. We and other groups demonstrated that the embryonic cortex of Ts1Cje mice is thinner than that of wild-type mice [5,27], and neurogenesis in the cerebral cortex was impaired in Ts1Cje embryos at embryonic day 14.5 (E14.5) [5,28]. Conversely, normal cortical neurogenesis in the Ts1Cje embryos at E15.5 and normal cognition of Ts1Cje adults in the Morris water maze test have been reported [29]. Recently, a Cre-loxP-based method to introduce defined chromosomal duplications into the Mmu16 was established, resulting in a number of mouse models being developed: Ts1Rhr, Yey series, Yah series, and Tyb series [30] (Figure 1). Although Dp(16)1Yey/+ (Dp16) mice carry an extra copy of the complete Hsa21 syntenic region on Mmu16 [31], Dp16 embryos reportedly show a normal brain size and normal cortical neurogenesis [32]. These observations suggest a possibility that the defects of prenatal neurogenesis might be caused by an extra chromatid (the amplified developmental instability hypothesis). This hypothesis is that most DS phenotypes are a result of a nongene-specific disturbance in chromosomal balance, leading to disrupted homeostasis [8]. However, it is true that some candidate genes that contribute to disturbed prenatal neurogenesis have been identified, suggesting the involvement of “gene dosage effects”, with most DS phenotypes affected at least in part by the overexpression of specific genes in Hsa21 in the impaired neurogenesis of the embryonic cortex of DS.

## 4. Candidate Genes Related to the Impaired Embryonic Neurogenesis in DS

### 4.1. Dual Specificity Tyrosine-Phosphorylation-Regulated Kinase 1A (Dyrk1a) and Regulator of Calcineurin 1 (Rcan1)

DYRK1A is considered to be involved in the neurodevelopment of DS [33,34] (Figure 2). iPSCs with trisomy 21 show abnormal neural differentiation that is largely improved by inhibiting or knocking down DYRK1A [18]. In embryonic cortical neurogenesis, the importance of cell cycle regulation has been suggested [35,36]. An extra copy of the *Dyrk1a* gene impairs the proliferation and G1 cell cycle duration in DS fibroblasts through direct phosphorylation and degradation of cyclin D1 [37]. Furthermore, the genetic normalization of the *Dyrk1A* gene dosage in Ts65Dn embryos by crossing Dyrk1a^+/−^ mice with Ts65Dn mice restores cyclin D1 to normal levels, an effect accompanied by the restoration of the number of cortical neurons during postnatal life [38].

A recent pharmacotherapeutic study involving maternal treatment with a DYRK1A inhibitor in Ts1Cje mice revealed that the inhibition of DYRK1A activity is sufficient to restore defects in cortical development in Ts1Cje mice [27]. Taken together, these findings suggest that an increased dosage of the *Dyrk1a* gene mediates the defects of early cortical neurogenesis in DS.

In addition to the *Dyrk1a* gene, the *Rcan1* gene, also known as *Dscr1* coded on Hsa21, was suggested to be involved in the impairment of embryonic cortical neurogenesis in DS (Figure 2). Both RCAN1 and DYRK1A cooperatively regulate the activity of nuclear factor of activated T-cell cytoplasmic (NFATc), which consists of isoforms NFATc1-4 [39]. RCAN1 inhibits the calcineurin-dependent signaling by associating with calcineurin A. Calcineurin A, a calcium and calmodulin-dependent serine/threonine protein phosphatase, activates NFATc via dephosphorylation. In contrast, DYRK1A phosphorylates NFATc4 and promotes NFATc4 export from the nucleus. Kurabayashi and Sanada demonstrated that NFATc suppression induced by the increased expression of RCAN1 and DYRK1A impaired cortical neurogenesis in Ts1Cje embryos [28].

RCAN1 overexpression affects the function of mitochondrial permeability transition pore (mPTP), resulting in impaired calcium retention, mitochondrial swelling and rupture of the outer membrane [40]. Mitochondrial dysfunction is suggested to lead decrease of embryonic neurogenesis [41,42]. In fact, fibroblasts from DS fetus and the brain of Ts1Cje mice harboring triplicated *Rcan1* gene showed also swelled mitochondria with damaged membranes [43,44]. Thus, *Dyrk1a* and *Rcan1* genes are promising candidates for causing altered brain development in DS fetuses.

### 4.2. Amyloid Precursor Protein (App)

A study using cultured NPCs obtained from Ts65Dn neonates revealed an increased production of the APP intracellular domain, which upregulates the expression of the putative receptor for sonic hedgehog (SHH) protein (PTCH1) and downregulates the expression of the SHH homolog pathway, resulting in the reduced proliferation of NPCs in the hippocampus and SVZ [45]. Overexpression of APP reduced the expression of miR-574-5p, which inhibits the proliferation of NPC proliferation, in the NPCs of developing cerebral cortex from E14.5 to neonatal day [46].

### 4.3. Down Syndrome Cell-Adhesion Molecule (DSCAM)

DSCAM is suggested to be associated with neuronal generation, maturation, and neuronal wiring [47,48,49,50,51,52,53,54,55,56]. Loss of *Dscam* gene in mice results in transient decrease in cortical thickness without an increase in cell death or reduction in progenitor proliferation during embryonic life [52]. In contrast, knockdown of DSCAM or DSCAM-like 1 (DSCAML1) in the cortex impairs the radial migration of projection neurons [57].

Recently, the involvement of the DSCAM/PAK1 pathway in neurogenesis deficits was demonstrated using DS iPSC-derived cerebral organoids [20]. Triplication of the Dscam gene disturbed the regulation of p21-activated kinase 1 (PAK1) activity, resulting in neuronal dysconnectivity in immortalized cells from trisomy 16 (Ts16) mouse embryos [58]. PAK1 is also suggested to be involved in cortical development through regulating the proliferation of NPCs [59]. Tang et al. showed that the reduced proliferation of NPCs resulted in the reduced size and expansion rates of DS iPSC-derived organoids, suggesting impaired neurogenesis in trisomy 21 organoids [20]. They also demonstrated that knocking down DSCAM and performing treatment with an inhibitor of PAK1 activation improved the proliferation deficits in DS organoids [20].

### 4.4. Oligodendrocyte Transcription Factor 2 (Olig2)

OLIG2, a basic helix–loop–helix transcription factor plays a role in the development of the mammalian central nervous system. OLIG2 is essential in oligodendrocyte development and formation of motor neurons from NPCs during embryogenesis [60,61,62]. OLIG2 is expressed in the specific boundaries of the brain regions, such as the hypothalamus and VZ and SVZ of the LGE and MGE at E13.0 [63,64]. Furthermore, the expression level of OLIG2 in the VZ progenitors is higher in the MGE, which is a region producing cortical interneurons from E9.5 to E16.5, than in the LGE [63,65]. OLIG2 overexpression is observed in the MGE of Ts65Dn [66] and Ts1Cje embryos [67], and Chakrabarti et al. showed that *Olig1* and *Olig2* triplication causes increased inhibitory neurogenesis in Ts65Dn mice by genetic normalization of the *Olig1*/*Olig2* dosage [66]. These findings indicate that normalization of the *Olig1* and *Olig2* gene copy number is sufficient to improve the overproduction of inhibitory neurogenesis in DS mouse models.

Mice overexpressing OLIG2 in cortical NPCs display microcephaly, cortical dyslamination, hippocampal malformation, and an impaired motor function [68]. In these transgenic mice, the cell cycle progression of cortical progenitors was also impaired, and a ChIP-seq analysis indicated that *Olig2* occupied the promoter or enhancer regions of *Nfatc4*, *Pax6*, *Dyrk1a*, and *Rcan1* genes that are related to reduced neurogenesis in DS [68].

OLIG2 abnormal expression in human iPSCs with trisomy 21 has been also reported. In ventral forebrain organoids generated from control and DS iPSCs, a significantly greater percentage of OLIG2-positive cells are detected in the organoids from DS iPSCs than those from control iPSCs [69]. Although the expression of Olig1 transcripts are significantly increased, OLIG1-positive cells observed are very few in number, and the OLIG1 protein levels in organoids derived from DS iPSCs are comparable to those in control organoids. These findings suggest a potential role of OLIG2 in neuronal differentiation in human cells at the early stage.

### 4.5. Expressed in Undifferentiated Retina and Lens of Chick Embryos (EURL/C21ORF91)

EURL, also known as C21ORF91, is coded at the centromeric boundary of the DS critical region (DSCR encompassing 21q21–21q22.3) and expressed in the cerebral cortex of mice and humans during brain development [70]. In addition, an elevated expression of EURL in the lymphoblastic cells derived from people with DS has been shown [71,72]. An individual with intellectual disability but lacking the typical clinical features of DS was found to have partial tetrasomy of Hsa21, including the *Eurl* gene, indicating that this Hsa21 region is associated with the development of DS intellectual disability [73]. Similarly, a male infant with microcephaly reportedly carried partial tetrasomy 21, including the *Eurl* gene [74]. These reports suggest that perturbations of the region in the vicinity of the EURL gene may be contributing genetic factors influencing the neurobiology of brain growth and intellectual disability. According to this hypothesis, Li et al. demonstrated that knockdown or overexpression of the *Eurl* gene affected the proliferation of neuroprogenitors and neuronal differentiation [70]. In addition, they also demonstrated the upregulation of neural β-catenin in response to the EURL overexpression. Wnt/β-catenin signaling is known to be a key regulator of oligodendrocyte development, as it is transiently activated in oligodendrocyte progenitor cells at the initiation of terminal differentiation [75]. In fact, EURL has been suggested to play a role in accurate oligodendroglial differentiation [76]. EURL overexpression further induces the generation of a cell population coexpressing both astroglial and oligodendroglial markers, indicating that elevated EURL levels induce a gliogenic shift towards the astrocytic lineage, reflecting non-equilibrated glial cell populations in brains with DS.

### 4.6. ETS Transcription Factor ERG

Defects of cortical neurogenesis are observed in Ts1Cje embryos at E14.5 [5]. To clarify the molecular alternations involved in this impaired neurogenesis, DNA microarray-based gene expression profiling analyses were performed, demonstrating that the expression of inflammation-related genes was dramatically upregulated in the embryonic brain of Ts1Cje mice [77].

The genetic normalization of the *Erg* gene dosage in Ts1Cje embryos by crossing Erg^+/mld2^ mice [78] and Ts1Cje mice restores the upregulation of inflammation-related gene expression. In addition, Ts1Cje-Erg^+/+/mld2^ embryos show normal cortical neurogenesis, suggesting that triplication of the Erg gene is involved in defective cortical neurogenesis in Ts1Cje mice [77]. ERG is involved in the development of both hematopoietic and endothelial cells. ERG has a critical function in normal hematopoiesis [78,79]. ERG is constitutively expressed in normal endothelial cells and regulates angiogenesis and endothelial apoptosis [80,81,82]. Angiopoietin-2, which plays a key role in new blood vessel formation, regulates cortical neurogenesis in the embryonic cortex [83]. Since angiogenesis therefore seems to affect neurogenesis in the developing brain, the increased expression of ERG in endothelial cells may affect cortical neurogenesis by inhibiting the cortical angiogenesis.

## 5. Conclusions and Perspectives

Accumulating evidence from human and mouse models indicates that prenatal neurogenesis for the formation of the cerebral cortex is impaired in DS, possibly resulting in delayed brain development. Several genes related to the impaired neurogenesis in DS have been identified through analyses with mouse models of DS. Of note, the relationships among these genes, except for *Rcan1* and *Dyrk1a*, remain unclear. The relationships among the genes in the trisomic region, introduced here, are not clarifed in the disturbed prenatal neurogenesis, although further evidence will need to be collected in the future. For example, whether or not triplication of the *Erg* gene is solely sufficient to impair the cortical neurogenesis is unclear. Similarly, whether or not triplication of both *Rcan1* and *Dyrk1a* genes is necessary to disturb the embryonic cortical neurogenesis is also unclear. Understanding the genetic relationships among the candidates introduced here may help explain the severity of cognitive impairment in DS mouse models. Since a number of mouse models—Dp(16)Yah, Dp(16)Yey, and Dp(16)Tyb mouse series—were established using modern methods of arranging chromosomes [30] (Figure 1), they may help clarify the genetic etiology of impaired neurogenesis in DS.

## Figures and Tables

**Figure 1 genes-12-01598-f001:**
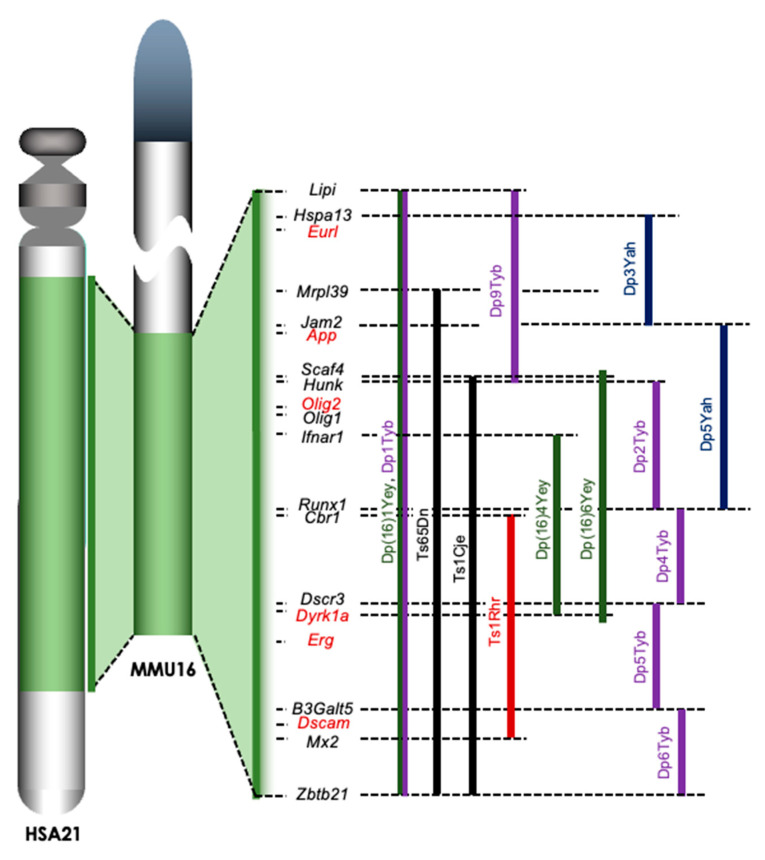
Trisomic regions of mouse models for DS. A large portion of Hsa21 is syntenic with the distal end of Mmu16. The trisomic regions in several mouse models of DS are compared on the right of Mmu16. Ts65Dn and Ts1Cje mice (shown in black) were established by accidental translocation of Mmu16 segments on Mmu17 and Mmu12, respectively. Ts1Rhr mice were the first model involving the engineered duplication (Dp) of DSCR (shown in red). New engineered Dp models have been developed in the last decade, including the Yey series (shown in green), Yah series (shown in dark blue), and Tyb series (shown in purple) established by Drs. Eugene Yu, Victor L. J. Tybulewicz, and Yann Herault, respectively. Lipi: lipase, member I, Hspa13: heat shock protein 70 family, member 13, Eurl: C21orf91 or D16Ertd472e, Mrpl39: mitochondrial ribosomal protein L39, Jam2: junction adhesion molecule 2, App: amyloid precursor protein, Scaf4: SR-related CTD-associated factor 4, Hunk: hormonally upregulated Neu-associated kinase, Olig1/Olig2: oligodendrocyte transcription factor 1/2, Ifnar1: interferon (α and β) receptor 1, Runx1: runt-related transcription factor 1, Cbr1: carbonyl reductase 1, Dscr3: Down syndrome critical region gene 3, Dyrk1a: dual-specificity tyrosine-(Y)-phosphorylation regulated kinase 1a, Erg: ETS transcription factor related gene, B3Galt5: UDP-Gal:betaGlcNAc β 1,3-galactosyltransferase, polypeptide 5, Dscam: DS cell adhesion molecule, Mx2: MX dynamin-like GTPase 2, and Zbtb21: zinc finger and BTB domain containing 21.

**Figure 2 genes-12-01598-f002:**
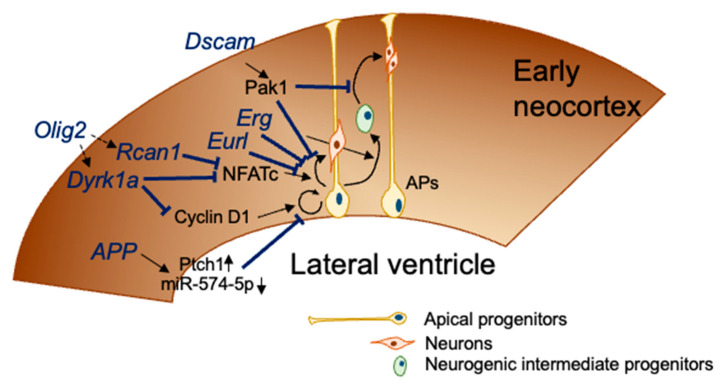
DS genes related to impaired embryonic cortical neurogenesis. At E12.5–E18.5, apical progenitors divide asymmetrically to self-renew and give rise to a neuron that migrates toward the cortical plate (direct neurogenesis) or to a neurogenic intermediate progenitor that migrates out of the ventricular zone to form the subventricular zone. Neurogenic intermediate progenitors divide to give rise to a couple of neurons. Seven genes coded in Mmu16, which is orthologous with Hsa21, disturb the neurogenesis process, as shown in this figure.

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
