# Peer review of "Genes Associated with Disturbed Cerebral Neurogenesis in the Embryonic Brain of Mouse Models of Down Syndrome"

_genes, 2021, doi:10.3390/genes12101598_

Round 1

Reviewer 1 Report

The manuscript clearly and concisely summarize the main aspects of the field. The title faithfully reflects the content of the review. The work is well organized and comprehensively written. Therefore, it is suitable for a wide readership: from a researcher in neurodevelopment who wants a quick overview of DS neurogenesis, to a wider audience interested in the neurobiology of chromosome 21 aneuploidies.

To conclude, the author hypothesizes that defects in embryonic cortical neurogenesis may lead to delayed brain development. In my opinion, the data presented in the manuscript do not constitute an irrefutable proof of this hypothesis and might also support the opposite one: a general delay in brain development could contribute to an altered cortical development. For this reason I would tone down that statement.

I also propose some specific comments in the attached document.

Author Response

Responses to Reviewers' comments:

I thank all of the reviewers for their useful comments and suggestions, and I have revised the manuscript accordingly.

Reviewer 1

To conclude, the author hypothesizes that defects in embryonic cortical neurogenesis may lead to delayed brain development. In my opinion, the data presented in the manuscript do not constitute an irrefutable proof of this hypothesis and might also support the opposite one: a general delay in brain development could contribute to an altered cortical development. For this reason I would tone down that statement.

Response: I agree with the reviewer’s opinion. That’s why I have revised as following:

“Accumulating evidence from human and mouse models indicates that prenatal neurogenesis for the formation of the cerebral cortex is impaired in DS, possibly resulting in delayed brain development.” (conclusion section)

-Line 42: “Furthermore, the neural progenitor cells (NPCs) in DS are suggested to show a greater  shift  towards  glial  lineages:  differentiating  into  microglia,  astrocytes,  and oligodendrocyte”. This sentence should be clarified, because it suggest that microglial cells are originated from neural progenitor cells. Kanaumi et al. (2013) shows an increases rate in the density of microglia through in the germinal matrix of DS fetuses. However, this work do not demonstrate that microglia is produced by NSC. In fact, microglial cells are the resident macrophages of the central nervous system and they do not originate from cortical NSC but from yolk sac progenitors that invades the mouse cerebral cortex around embryonic day 10.5 (E10.5; Swinnen N, Smolders S, Avila A, et al., 2012, Complex Invasion Pattern of  the  Cerebral  Cortex  by  Microglial  Cells  During  Development  of  the  Mouse  Embryo, Glia:  61,  150-163;  Tong  CH  and  Vidyadaran  S,  2016,  Role  of  microglia  in  embryonic neurogenesis, Exp Biol Med: 241, 1669-1675).

Response: I agree with the important revirwer’s suggestion. I have corrected it as following:

“Furthermore, fewer neurons and more astrocytes are found in the prenatal brain with DS, suggesting that the neural progenitor cells (NPCs) in DS show a greater shift towards glial lineages: differentiating into astrocytes, preferentially [4,13,14].”

-Line 125: : ”An extra copy of the Dyrk1a gene confers impairs the proliferation and G1 cell cycle duration in DS fibroblasts through direct phosphorylation and degradation of cyclin D1”. There are two verbs in this sentence. I propose to eliminate “confers”.

Response: Thank you for your suggestion. I have corrected this sentence as suggested.

-Line 145: “Thus, Dyrk1a and Rcan1 genes are promising candidates for causing delayed brain development in DS fetuses”. In the developing cerebral cortex, the overexpression of Dyrk1a cause an early production of intermediate progenitors and oligodendrocyte precursor cells (OPCs) from cortical NPCs (Najas S, Arranz J, Lochhead PA, et al., 2015, DYRK1A-mediated Cyclin D1 degradation in neural stem cells contributes to the neurogenic cortical defects in Down syndrome. EbioMedicine: 2, 120-134). It does not support that Dyrk1a causes a developmental delay but an advance in the developmental program of NPCs. For this reason I propose to change “delayed” by “altered”. However, if the author refers to the action of Dyrk1a and Rcan together, it might be emphasized to make it absolutely clear.

Response: Thank you so much for your suggestion. I have substituted “delayed” with “altered” according to the suggestion.

-References: The manuscript contains appropriate and adequate references. However I miss the following two references reporting a reduction in the number of neurons in DS cerebral cortex at birth and in the adulthood (Ross et al., 1984; Schmidt-Sidor et al., 1990), and proposing that DS fetuses exhibits a prenatal retardation of neurogenesis at 22 weeks of gestational age (Schmidt-Sidor et al., 1990).

Ross MH, Galaburda AM, and Kemper TL. 1984. Down`s syndrome: Is there a decreased population of neurons?, Neurology, 34, 909-916.

Schmidt-Sidor B, Wisniewski KE, Shepard TH, and Sersen EA. 1990. Brain growth in Down syndrome subjects 15 to 22 weeks of gestational age and birth to 60 months. Clinical Neuropathology, 4: 181 -190.

Response: Thank you for the valuable information. I have added these references in the revised manuscript.

Reviewer 2 Report

Trisomy of chromosome 21 (TS21) is the most common autosomal aneuploidy compatible with postnatal survival .Its phenotype is highly complex with constant features, such as mental retardation,.   The mechanism by which this aneuploidy produces the clinical phenotype and the phenotypic variations are largely unknown. Many hypotheses have been proposed that DS phenotypes results from the dosage imbalance of multiple genes.   This review summarize how this overexpression of genes on chromosome-21 is responsible for the altered neuronal development.   Overexpression of APP induces mitochondrial oxidative stress and activates the intrinsic apoptotic cascade. Interestingly, RCAN1 overexpression has been linked to oxidative stress and mitochondrial dysfunction. Mitochondrial dysfunction plays a primary role in neurodevelopmental anomalies and neurodegeneration of Down syndrome (DS) subjects. For this reason, targeting mitochondrial key genes, such as PGC-1α/PPARGC1A, is emerging as a good therapeutic approach to attenuate cognitive disability in DS.   DYRK1A and DSCR1/RCAN1, have been proposed as possible candidates for mitochondrial abnormalities as they control PGC-1α via the calcineurin/NFAT pathway   Maybe you could integrate this in your review .   Even add more figures could help to understand the concepts   The review is quite simply and clear   Kind regards

Author Response

Responses to Reviewers' comments:

I thank all of the reviewers for their useful comments and suggestions, and I have revised the manuscript accordingly.

Reviewer 2

Trisomy of chromosome 21 (TS21) is the most common autosomal aneuploidy compatible with postnatal survival .  Its phenotype is highly complex with constant features, such as mental retardation,.   The mechanism by which this aneuploidy produces the clinical phenotype and the phenotypic variations are largely unknown. Many hypotheses have been proposed that DS phenotypes results from the dosage imbalance of multiple genes.   This review summarize how this overexpression of genes on chromosome-21 is responsible for the altered neuronal development.   Overexpression of APP induces mitochondrial oxidative stress and activates the intrinsic apoptotic cascade. Interestingly, RCAN1 overexpression has been linked to oxidative stress and mitochondrial dysfunction. Mitochondrial dysfunction plays a primary role in neurodevelopmental anomalies and neurodegeneration of Down syndrome (DS) subjects. For this reason, targeting mitochondrial key genes, such as PGC-1α/PPARGC1A, is emerging as a good therapeutic approach to attenuate cognitive disability in DS.   DYRK1A and DSCR1/RCAN1, have been proposed as possible candidates for mitochondrial abnormalities as they control PGC-1α via the calcineurin/NFAT pathway   Maybe you could integrate this in your review .   Even add more figures could help to understand the concepts   The review is quite simply and clear 

Response: Thank you so much for your reviewing. I have added your valuable proposal in the revised manuscript as following:

RCAN1 overexpression affects the function of mitochondrial permeability transition pore (mPTP), resulting in impaired calcium retention, mitochondrial swelling and rupture of the outer membrane [40]. Mitochondrial dysfunction is suggested to lead decrease of embryonic neurogenesis [41,42]. In fact, fibroblasts from DS fetus and the brain of Ts1Cje mice harboring triplicated Rcan1 gene showed also swelled mitochondria with damaged membranes [43,44].